# Trends in Diabetes Prevalence, Awareness, Treatment and Control in Yangon Region, Myanmar, Between 2004 and 2014, Two Cross-Sectional Studies

**DOI:** 10.3390/ijerph16183461

**Published:** 2019-09-18

**Authors:** Wai Phyo Aung, Espen Bjertness, Aung Soe Htet, Hein Stigum, Marte Karoline Råberg Kjøllesdal

**Affiliations:** 1Department of Community Medicine and Global Health, Institute of Health and Society, Faculty of Medicine, University of Oslo, 0318 Oslo, Norway; espen.bjertness@medisin.uio.no (E.B.); aungsh@gmail.com (A.S.H.); hein.stigum@medisin.uio.no (H.S.); 2Procurement and Supply Division, Department of Public Health, Ministry of Health and Sports, Nay Pyi Taw 15011, Myanmar; 3International Relations Division, Ministry of Health and Sports, Nay Pyi Taw 15011, Myanmar; 4Cluster for Health Services Research, Norwegian Institute of Public Health, 0213 Oslo, Norway; MarteKarolineRaberg.Kjollesdal@fhi.no

**Keywords:** diabetes mellitus, prevalence, awareness, treatment, control, Yangon, Myanmar

## Abstract

Myanmar is currently facing the burden of non-communicable diseases due to changes in lifestyle and dietary patterns linked to socio-economic development. However, evidence is scarce about changes in the prevalence of diabetes mellitus (DM) over time. We aimed to investigate changes in the prevalence, awareness, treatment and control of DM from 2004 to 2014, among adults aged 25–74 years, in the Yangon region. Two cross-sectional household-based studies, according to World Health Organization STEPwise approach to surveillance (WHO STEPS) methodology, were conducted in 2004 (*n* = 4448) and 2014 (*n* = 1372). The overall age-standardized prevalence of DM was 8.3% (95% CI 6.5–10.6) in 2004 and 10.2% (7.6–13.6) in 2014 (*p* = 0.296). The DM prevalence increased between the study years among elderly participants only, from 14.6% (11.7–18.1) to 31.9% (21.1–45.0) (*p* = 0.009). Awareness of having DM increased from 44.3% (39.2, 49.6) to 69.4% (62.9–75.2) (*p* < 0.001). Among participants who were aware of having DM, the proportion under treatment increased from 55.1% (46.8–63.1) to 68.6% (61.5–74.8) (*p* = 0.015). There was no change in proportion with controlled DM. Adjusted for age, sex and education, mean fasting plasma glucose levels in 2014 were 0.56 mmol/L (0.26–0.84) higher than in 2004. Preventive measures to halt future increases in DM prevalence and to increase the detection of undiagnosed DM cases are needed.

## 1. Introduction

Diabetes Mellitus (DM) has become a global epidemic partly due to a growing and longer-living population [1,2]. The age-standardized prevalence of DM increased substantially from 1980 to 2014, from 4.3% to 9.0% in men and from 5.0% to 7.9% in women [3]. The International Diabetes Federation (IDF) estimated that 327 million people globally lived with DM in 2017 and the number is predicted to increase to 438 million by 2045 [1]. High fasting plasma glucose (FPG) was one of ten major contributors to disability adjusted life years (DALYs) worldwide in 2015 [4], and will, if not well controlled, lead to long-term complications such as cardiovascular diseases (CVD), end-stage renal disease, lower limb amputation, and retinopathy [5]. Globally, health care expenditure related to DM treatment and consequences is increasing [1]. The best way to halt the rising burden is to improve primary prevention and early detection, especially in developing countries [6]. Consequently, the United Nations has declared a one-third reduction of premature deaths caused by non-communicable diseases (NCD) by 2030 as part of the sustainable development goals [7]. In the South-East Asia Region (SEAR), DM has increasingly contributed to DALYs from 1990 to 2017 (from 1.9% of total DALYs to 5.0%) [8].

In the Western Pacific region, including Myanmar, the number of people suffering from DM is estimated to increase by 15% between 2017 and 2045, according to IDF [1]. As a consequence of socio-economic development in Myanmar, accompanied by changes in lifestyle including dietary patterns [9] and more sedentary lifestyles [10,11], the country is facing the burden of NCD [12]. The contribution of DM to DALYs in Myanmar increased from 2.3% to 5.5% between 1990 and 2017 [8]. In addition, the country has a high burden of communicable diseases [13]. The health care system is facing challenges in addressing the rise in DM care, such as the supplementation of essential drugs and equipment for diabetes care, capacity building of endocrinologists, inequality of DM care between urban and rural areas, an inappropriate referral system in DM care and health information system and lack of dieticians and physical activity instructors [14]. Data on DM prevalence in Myanmar is scarce. In 2014, a national STEPS survey was conducted, which showed that the prevalence of DM in Myanmar was 10.5% in total, 9.1% in male and 11.8% in female [10]. In 2003–2004, a sub-national survey (Yangon Region) was conducted as the first STEPS survey in Myanmar [15], which was repeated in 2014, both assessing DM prevalence and related outcomes. Whether there have been any changes over time in the awareness, treatment and control status of DM in Myanmar is not known. Therefore, the aim of this article was to assess changes in the prevalence, awareness, treatment and control of DM from 2004 to 2014, among adults aged 25–74 years, in the Yangon region of Myanmar.

## 2. Materials and Methods

### 2.1. Methods

We analyzed the data from two cross-sectional studies in urban and rural areas of the Yangon Region, in 2004 and in 2014 [16,17]. Both studies followed the WHO STEPS methodology, including 3 STEPS [18]. STEP 1 included questionnaires covering socio-demographic background, behavioral pattern and history of diabetes and hypertension of the participants. STEP 2 included anthropometric and blood pressure measurements. STEP 3 was a biochemical investigation including FPG and fasting lipid profiles (total cholesterol (TC), triglycerides (TG)).

Men and women aged 25–74 years old were eligible for participation in the study. Military personnel, institutionalized persons, monks and nuns and individuals too physically or mentally ill to participate were excluded from the study. Using the WHO STEPS sample size calculator, with a response rate of 80%, marginal error of 5%, confidence interval of 95%, and the design effect of 1.5, a sample size of 5000 was deemed sufficient in 2004, and 1600 in 2014. Pregnant women were excluded as physiological changes and gestational diabetes might affect the estimates. The number of participants was 4448 in the 2004 study and 1486 in the 2014 study. The response rate of STEPs 1 and 2 were 89% and 92% in the 2004 and 2014 study, respectively, while the response rate of all completed STEPs were 89% in the 2004 study and 86% in the 2014 study. The reasons given for not participating in STEPs 1 and 2 were ‘not willing’ and ‘not having time’. In STEP 3, non-response was due to worries about blood tests.

### 2.2. Sampling

Based on multistage cluster sampling for both studies, urban and rural settings were first identified in the Yangon Region. Second, 10 townships from urban areas and 5 townships from rural areas were selected randomly in 2004, whereas in 2014, six townships from urban and six from rural areas were selected randomly. Third, 10 wards from urban areas and 10 villages from rural areas were randomly selected from each of the selected townships in 2004 while 5 wards and 5 villages were similarly randomly selected in 2014. Furthermore, 25–27 households were selected randomly from selected units. Gender was stratified in this stage for the 2014 study only. Finally, one eligible respondent was randomly selected from each selected household.

### 2.3. Data Collection and Measurement

In accordance with the STEPS methodology [18], training of interviewers and practice sections including pilot surveys were conducted in the same manner for both studies. We used the STEPS instruments version 1.3 for the 2004 study while version 2.1 was used in 2014.

Fasting venous blood samples were collected after overnight fasting for at least 10 h at a survey site or health facility by health professionals for both studies. We used the glucose tube containing fluoride. Cold boxes with ice were prepared for the storage of blood tubes at the blood sample collection site. Then, blood samples were transported to the laboratory of the Department of Medical Research (Lower Myanmar), Yangon in 2004 and to the National Health Laboratory, Yangon in 2014. An enzymatic colorimetric test was used for FPG assessment in 2004, while an enzymatic reference method with haxokinase was used for FPG assessment in 2014. The reagent of Human Gesellschaft fur Biochemica und Diagnostica mbh, Germany was used in 2004 and that of Roche Diagnostics, Indianapolis, USA in 2014.

### 2.4. Variables

DM was defined as FPG levels of ≥ 7 mmol/L and/or self-reported DM as diagnosed by health professionals [19]. Awareness of DM was defined as being aware of having been diagnosed with DM by a medical doctor or other health care personnel, among those defined as having DM. The treatment of DM was defined as taking treatment for DM in terms of insulin and/or oral hypoglycemic agents. Control of DM was defined as FPG level < 7 mmol/L among participants on DM treatment.

Education level was defined as total number of school years and classified into “no formal education” (0 year), “primary education” (1–5 years), “secondary education” (6–11 years) and higher education ( ≥ 12 years). We defined daily income as the household income earned by all household members. It was converted from local currency (Myanmar Kyats) to United States Dollar (USD). With an exchange rate of 1 USD = 750 Kyats in 2004 [20] and 953.8 Kyats in 2014 [21]. We used the international poverty line of USD 1.9/day, defined by the World Bank for daily income subgroups [22].

### 2.5. Statistical Methods

Based on the age and gender distribution of the Yangon Region, specified weightage of different levels of sampling units on the study population according to the Myanmar census 2014 report [23] were considered. Prevalence, awareness, treatment, control of DM and mean FPG estimates in both studies were age-standardized in accordance with the Myanmar census 2014 [23] because of expected differences in age distributions between 2004 and 2014. We used the ‘svy’ command by declaring the complex survey design for the estimation of mean, proportion and in linear and logistic regression analyses. Chi-square tests and Wald test were used to assess differences between 2004 and 2014 in socio-demographic background variables and in mean FPG levels. Multiple linear regression analyses were conducted to assess the changes of the FPG level between 2004 and 2014, and multiple logistic regressions were also performed for estimating the changes of DM prevalence during this period, adjusted for potential confounders. We used a directed acyclic graph (DAG) [24] to identify possible confounders of the relationship between study year and FPG/DM. Age, sex and education were defined as confounders as they were associated with both selection into the study and with the outcome, and were thus adjusted for in regression analyses. Age was used as a continuous variable in the regression analysis. Those with missing data on income (*n* = 277 in 2004, *n* = 79 in 2014) were included in analyses not relevant to income. A *p* value, < 0.05 was regarded as statistical significance. We used STATA/IC version 15.1 for analyses of the data.

## 3. Results

The distribution of urban-rural location and gender was comparable between the two studies (Table 1). A higher proportion of the participants in 2014 compared to 2004 were in the highest educational group and had an income of ≥ 3.1 USD/day.

The overall age-standardized DM prevalence did not change from 2004 to 2014 (Table 2). However, an increase in prevalence was seen among those aged ≥ 60 years from 14.6% to 31.9% (*p* = 0.009) (data not shown in the table). The proportion of the participants who were aware of having DM increased between 2004 and 2014 among women, from 45.2% (95% CI 38.8–58.8) to 76.4% (69.9, 81.9), but not among men. Among participants who were aware of having DM, the proportion under treatment significantly increased between 2004 and 2014 among men from 54.1% (47.1, 61.0) to 81.8% (78.2, 85.0), and among urban women from 54.9% (46.9, 62.7) to 69.6% (66.7, 72.2). Among participants under treatment, the proportion with controlled DM had not increased from 2004 to 2014.

The mean FPG levels increased from 5.14 (SE) ± 0.09 in 2004 to 5.59 (SE) ± 0.06 in 2014 (Table 3). Increases in FPG levels were seen in both rural and urban areas, and among women and men, except among urban women.

The mean FPG level in 2014 was 0.49 mmol/L (0.20, 0.78) higher than that of 2004 in the crude model (Table 4). After adjustment for age, sex and education, the estimate changed to 0.56 mmol/L (0.26, 0.84). There was no significant increase in odds for DM in 2014 as compared with 2004 (Table 5).

## 4. Discussion

Mean plasma glucose levels were higher in 2014 than in 2004, but an increase in DM prevalence during this period was seen only among the oldest participants. The proportion of women who were aware of having DM, as well as the proportion under treatment for most groups increased, however not the proportion in the control group.

### 4.1. Prevalence of Diabetes Mellitus

The overall prevalence of DM in the Yangon region in 2004 and 2014 were similar to what was reported in Thailand in the same period (7.7% in 2004, 9.9% in 2014) [25], and also in Korea between 2001 (9.2% among men, 7.7% among women) and 2009 (11.0% among men, 9.2% among women) [26]. As Thailand and Korea had more developed economies than Myanmar in 2004, the prevalence in Myanmar at this time was notably high. The increase over the 10-year period was statistically significant in the other two countries, but not in our study. Myanmar started its economic reform process from 2011 after changing to a democratic government, and the effect of this process on the prevalence of DM may not have been fully captured within the period of our study. In Malaysia, a longer study period was needed to register changes in DM prevalence following improvements in economy [27]. The educational level in our study was higher in 2014 than in 2004, following economic reforms in the country, and we may speculate that the reform has mitigated an increased prevalence of DM following recent lifestyle changes.

The level of awareness of DM has been rising. This is consistent with the effort of the National Health Plan (2006–2011), entailing components to raise awareness of DM and to promote a healthy lifestyle through changes in environmental factors, as well as early detection of DM [28]. However, despite these efforts, a study from 2013–2014 showed a high prevalence of risk factors for NCDs in the Yangon region, with metabolic risk factors such as being overweight, obesity and diabetes being most prevalent in urban areas, and behavioral risk factors, such as alcohol consumption and low fruit and vegetable intake in rural areas [17]. Furthermore, we saw an increase in mean levels of plasma glucose from 2004 to 2014, suggesting that a rise in DM prevalence will occur in the years to come. Quality orientated health systems, supporting programs for self-management for the patients, as well as developing a culture for healthy lifestyles has been recommended for simultaneous implementation to halt increases in DM prevalence and the burden it poses to health systems [29].

The increasing trend of DM in the elderly population is in line with findings from China [30]. A substantial increase of elderly people in the population in the coming years further increases the probability of a more pronounced increase in DM prevalence in the Yangon region in the years to come [31,32]. In Singapore, a decreasing trend of DM prevalence (from 10.0% in 1992 to 7.8% in 2004) [33] was recorded following achievements of the national healthy lifestyle campaign launched in 1992, which included a series of health promotion measures. It used mass media to deliver intensive health education regarding a healthy lifestyle, including choosing healthy food, regular physical activity, and the disadvantages of smoking and alcohol consumption [34]. Thus, such programmes could also be implemented in Myanmar.

### 4.2. Awareness, Treatment and Control

The levels of awareness and treatment of DM has increased significantly during the last decade in the Yangon region, however, not the levels of controlled DM. The proportion under treatment were higher among men than women, although a higher proportion of women compared to men were aware of having DM. One possible explanation is that women to a larger degree than men might be treated by herbal or traditional medicine, which was not defined as a treatment in our study. This is supported by findings of the National STEPS Survey of 2014 in Myanmar in which the proportion of consumption of herbal medicine was 32% among women and 14% among men [10]. Additionally, in the 2014 national survey, the proportion under treatment for DM was lower among women than among men, 74% and 81% for oral hypoglycemic agent treatment and 6% and 14% for insulin treatment, respectively [10]. Moreover, another study in Myanmar reported that many DM patients prefer Myanmar traditional medicine due to its accessibility and affordability. Regarding controlled DM, the prevalence in our study was similar to Thailand estimates (2004–2014), with a larger proportion of women than men with controlled DM in 2004, but with reversed proportions among men and women in 2014 [25].

### 4.3. Urban-Rural Differences

In the healthcare setting of Myanmar, urban inhabitants can easily consult a general practitioner for their health [9] while rural dwellers rely on health assistants and midwives [14]. Different access to quality care and health information [35] might lead to differences in awareness and treatment levels between urban and rural inhabitants. The proportion of men under treatment increased from 2004 to 2014 in both rural and urban areas, but only in urban areas among women. Traditional gender roles in rural areas could be an explanation for the lack of increase in treatment among rural women. Men are the breadwinners and primary decision makers in the family, while women spend their time caring for family (such as child raising), and might have fewer opportunities to care for their own health and visit health facilities [36]. Lower income among women than men could also be of importance for the gender difference in rural areas [37].

Despite better access to quality care in urban compared to rural areas, no increase in controlled DM between the study years was seen in either area. Unsatisfactory levels of availability and accessibility to health care services, and the lack of population-based initiatives for health promotion [38] could contribute to low levels of controlled DM in Myanmar. Health insurance is not established in Myanmar, and diabetes patients often experience large financial problems while they use up their pocket money for their long-term blood glucose monitoring and treatment [9], leading to failure in regular follow-up and monitoring. Low health care expenditure has been shown to lead to poor coverage of health facilities and poor health outcomes [39].

The strengths of the study include use of the standardized STEPS protocol for both studies. Both studies had a high response rate. Exclusion of military personnel, institutionalized persons, monks and nuns was done as these people potentially have a lifestyle different from the majority of the population. It cannot be ruled out whether the exclusions have led to a slight over or under estimation of the results. Although different townships were included in the 2004 and 2014 study, we used random sampling to get representative samples and had a large sample size. DM was defined based on FPG and self-report in both studies to be able to compare the prevalence between the two studies [35], as oral glucose tolerance test (OGTT) was performed in 2004 only, and not in 2014. This might have led to an underestimation of the results [19]. To check the influence of this, we estimated the age-standardized prevalence of DM in 2004, including OGTT, but it was not substantially different from what we reported (8.3%). Thus, it is likely that our results have captured the majority of DM cases in the Yangon region in 2004 and 2014. Moreover, in studies with comparable DM prevalence from Thailand [25] and Korea [26], only FPG was used for the assessment of trends in DM.

## 5. Conclusions

During the period between 2004 and 2014, mean plasma glucose levels increased in the Yangon region. However, the DM prevalence increased only among elderly participants. As we probably did not fully capture the effect of the economic reform in Myanmar, and the prevalence of risk factors for DM is suggested to be on the rise, the studies to monitor the development of DM prevalence in Myanmar are needed. Studies on the prevalence of DM and factors associated with risk will enable policy makers to initiate timely and efficient preventive measures. Moreover, measures to increase the detection rate of undiagnosed DM cases, treatment rate and also to help people with DM to control their situation in all primary, secondary and tertiary level are needed. This study suggests a focus on the management of diabetes, including the use of traditional medicine, and on gender differences in future research.

## Figures and Tables

**Table 1 ijerph-16-03461-t001:** Socio-demographic characteristics of 25–74-year-old citizens in the Yangon Region, Myanmar by study year and gender.

Socio-Demographic Characteristics	2004		2004 (*n* = 4448) N (%)	2014		2014 (*n* = 1372) N (%)	*p* Value *
	Male	Female	Total	Male	Female	Total	
Age group							0.000
25–34	320 (16.1)	387 (15.8)	707 (15.9)	125 (18.4)	134 (19.4)	259 (18.9)	
35–44	389 (19.5)	556 (22.7)	945 (21.3)	141 (20.7)	175 (25.3)	316 (23.0)	
45–54	492 (24.7)	639 (26.0)	1131 (25.4)	156 (22.9)	171 (24.8)	327 (23.8)	
55–64	442 (22.2)	539 (22.0)	981 (22.1)	162 (23.8)	144 (20.8)	306 (22.3)	
65–74	352 (17.6)	333 (13.6)	684 (15.4)	97 (14.2)	67 (9.7)	164 (12.0)	
Location							0.577
Urban	979 (49.1)	1306 (53.2)	2285 (51.4)	339 (49.8)	354 (51.2)	693 (50.5)	
Rural	1015 (50.9)	1148 (46.8)	2163 (48.6)	342 (50.2)	337 (48.8)	679 (49.5)	
Education level							0.000
No formal education	99 (5.0)	255 (10.4)	354 (8.0)	42 (6.2)	44 (6.4)	86 (6.3)	
Primary education	1063 (53.3)	1325 (54.0)	2388 (53.7)	300 (44.1)	341 (49.4)	641 (46.7)	
Secondary education	667 (33.5)	691 (28.2)	1358 (30.5)	229 (33.6)	194 (28.1)	423 (30.8)	
Higher education	165 (8.3)	183 (7.5)	348 (7.8)	110 (16.2)	112 (16.2)	222 (16.2)	
Daily income (USD/day) ****			(*n* = 4171) **			(*n* = 1293) ***	0.000
Less than 1.9	1260 (63.2)	1509 (61.5)	2769 (62.3)	380 (55.8)	363 (52.5)	743 (54.2)	
1.9–3.09	361 (18.1)	510 (20.8)	871 (19.6)	123 (18.1)	129 (18.7)	252 (18.4)	
≥ 3.1	271 (13.6)	260 (10.6)	531 (11.9)	141 (20.7)	157 (22.7)	298 (21.7)	

* Chi-square test to compare 2004 and 2014 ** 277 missing values in 2004 and *** 79 missing value in 2014 for daily income variable. **** 1 USD = 750 Kyats in 2004 and 953.8 Kyats in 2014.

**Table 2 ijerph-16-03461-t002:** Age standardized prevalence, awareness, treatment and control of diabetes mellitus among 25–74-year-old citizens in 2004 and 2014, Yangon Region, Myanmar.

Diabetes Mellitus	Male			Female			Total		
	2004	2014	*p* Value	2004	2014	*p* Value	2004	2014	*p* Value
Prevalence									
Total	7.6 (5.7–10.0)	8.6 (5.5–13.0)	0.635	8.9 (6.7–11.7)	11.4 (8.6–14.8)	0.211	8.3 (6.5–10.6)	10.2 (7.6–13.6)	0.296
Urban	9.7 (7.0–13.4)	11.1 (7.2–16.7)	0.603	10.1 (7.1–14.4)	13.2 (9.8–17.7)	0.241	9.9 (7.4–13.3)	12.1 (8.4–17.0)	0.395
Rural	3.4 (1.5–7.9)	5.6 (3.5–8.8)	0.246	6.0 (3.7–9.6)	8.3 (5.6–12.3)	0.254	4.9 (3.2–7.6)	7.1 (5.7–8.8)	0.099
Awareness									
Total	39.7 (31.7–48.3)	38.0 (32.4–43.9)	0.728	45.2 (38.8–51.8)	76.4 (69.9–81.9)	0.000	44.3 (39.2–49.6)	69.4 (62.9–75.2)	0.000
Urban	41.4 (33.7–49.6)	52.0 (42.7–61.2)	0.086	44.3 (35.9–52.9)	75.2 (69.3–80.3)	0.000	42.1 (36.6–47.9)	74.1 (66.1–80.8)	0.000
Rural	35.3 (26.2–45.5)	31.1 (27.0–35.4)	0.393	43.8 (39.4–48.4)	68.8 (62.9–74.0)	0.000	42.2 (36.3–48.3)	51.9 (42.9–60.7)	0.075
Treatment									
Total	54.1 (47.1–61.0)	81.8 (78.2–85.0)	0.000	59.6 (52.2–66.6)	66.8 (59.9–73.1)	0.141	55.1 (46.8–63.1)	68.6 (61.5–74.8)	0.015
Urban	58.6 (51.8–64.1)	77.9 (73.1–82.0)	0.000	54.9 (46.9–62.7)	69.6 (66.7–72.2)	0.002	52.8 (43.0, 62.5)	70.6 (66.0–74.9)	0.003
Rural	37.5 (32.2–43.0)	94.2 (62.3–98.4)	0.000	66.6 (60.4–72.2)	66.0 (54.3–76.1)	0.917	59.5 (48.4–69.7)	77.6 (62.4–84.8)	0.049
Control									
Total	27.8 (14.6–46.5)	48.4 (31.6–46.5)	0.090	31.6 (19.3–41.7)	35.5 (28.9–42.7)	0.619	30.0 (21.6–40.2)	40.8 (32.5–49.5)	0.096
Urban	30.6 (15.8–50.7)	44.9 (27.7–63.4)	0.255	27.3 (13.4–47.6)	46.1 (40.9–51.3)	0.145	28.8 (18.8–41.2)	42.5 (33.2–52.5)	0.068
Rural	– *	–	–	46.2 (21.7–72.6)	20.8 (7.9–44.5)	0.121	36.4 (22.5–52.9)	35.2 (16.4–60.1)	0.927

* No rural men were in the control group in 2004.

**Table 3 ijerph-16-03461-t003:** Age-standardized mean fasting plasma glucose level among 25–74-year-old-citizens in Yangon Region, Myanmar in 2004 and 2014.

Mean Fasting Plasma Glucose Level			
	2004	2014	*p* Value *
	Mean (SE)	Mean (SE)	
Total	5.14 (0.09)	5.59 (0.06)	0.001
Sex			
Male	5.05 (0.08)	5.55 (0.08)	<0.001
Female	5.22 (0.11)	5.62 (0.06)	0.004
Location			
Urban			
Urban Total	5.35 (0.13)	5.72 (0.06)	0.020
Urban Male	5.30 (0.12)	5.69 (0.09)	0.020
Urban Female	5.41 (0.16)	5.69 (0.05)	0.117
Rural			
Rural Total	4.72 (0.05)	5.37 (0.09)	<0.001
Rural Male	4.61 (0.09)	5.31 (0.07)	<0.001
Rural Female	4.81 (0.07)	5.43 (0.11)	<0.001

* Walt Test.

**Table 4 ijerph-16-03461-t004:** The association between study year (2014 versus 2004) and mean fasting plasma glucose levels among 25–74 year old citizens, Yangon, Myanmar, from linear regression analysis.

Study Year	Crude Model	Model 1
	ß (95% CI)	ß (95% CI)
2004	0	0
2014	0.49 (0.20–0.78) **	0.56 (0.26–0.84) **

Model 1: adjusted for age, sex and education level; ** *p* value < 0.01.

**Table 5 ijerph-16-03461-t005:** Odds ratio of diabetes mellitus in study year 2014 compared with study year 2004, among Yangon citizens, Yangon, Myanmar, from logistic regression analysis.

Study Year	Crude Model	Model 1
	OR (95% CI)	OR (95% CI)
2004	1	1
2014	1.41 (0.85–2.33)	1.63 (0.95–2.81)

Model 1: adjusted for age, sex and education level.

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
