# Peer review of "Trends in Diabetes Prevalence, Awareness, Treatment and Control in Yangon Region, Myanmar, Between 2004 and 2014, Two Cross-Sectional Studies"

_ijerph, 2019, doi:10.3390/ijerph16183461_

Round 1
Reviewer 1 Report
The authors have performed a longitudinal study about diabetes trends between 2004-2014 in Myanmar.
My concerns:
1. The authors are kindly requested to conform their report to an appropriate statement of the Equator Network. Since no Statement exists about the reporting of longitudinal studies, they should adjust their report according to
Zavada J, Dixon WG, Askling J. Launch of a checklist for reporting longitudinal observational drug studies in rheumatology: a EULAR extension of STROBE guidelines based on experience from biologics registries. Ann Rheum Dis. 2014;73(3):628.
Author Response
Response to reviewer 1 comment
Dear reviewer,
Thank you for the valuable comment and suggestion to our article. Here is how we have responded to it.
Yours sincerely,
Wai Phyo Aung
Point 1: The authors have performed a longitudinal study about diabetes trends between 2004- 2014 in Myanmar.
My concerns:
The authors are kindly requested to conform their report to an appropriate statement of the Equator Network. Since no statement exists about the reporting of longitudinal studies, they should adjust their report according to
Zavada J, Dixon WG, Askling J. Launch of a checklist for reporting longitudinal observational drug studies in rheumatology: a EULAR extension of STROBE guidelines based on experience from biologics registries. Ann Rheum Dis.
2014;73(3):628.
Response 1: Thank you very much for your comment. Our studies are two cross-sectional studies that were conducted in 2004 and 2014, respectively. It is not a longitudinal study and also the participants recruited in both studies were different because our studies were followed the methodology of WHO STEPS methodology. Therefore, we are reporting STROBE Checklist for cross-sectional studies as an attachment file.
Please see the attachment for STROBE Checklist for cross-sectional studies.

Reviewer 2 Report
46 Spell out non-communicable diseases the first time it is used and bracket (NCD)s
54-5 As NCD is now to be spelled out in line 46, no need to do so here. Remove the bracket from (NCD).
73 Spell out fasting plasma glucose followed by bracketed (FPG).
200 Substitute “rise” for “raise”.
205 Make “increase” plural by adding an “s” at the end.
211 Add the article “the” before “disadvantages”.
216 Each of “male” and “female” in this line should be made plural by adding an “s” at the end.
220 “Male” and “female” in this line should be made plural by adding an “s” at the end.
234 Make “role” plural by adding an “s” at the end.
235 Add the article “the” before “breadwinners”.
236 Bracket (such as child raising). Substitute “fewer” for “less”.
250 Eliminate “-“.
253 Eliminate “,”.
268 The conclusion should also explain the value in doing the research and how it might be helpful for future research.
287 Remove the semi-colon before 2017.
333 Eliminate duplicated “2004”.
346 Eliminate duplicated “2018”.
347 Eliminate space before “Kim”.
352 Eliminate two spaces before “Ministry”.
373 Change hyphen to dash.
375 Eliminate space before “Latt”.
Author Response
Response to reviewer 2 comment
Dear reviewer,
Thank you for the valuable comments and questions to our article. Here is how we have responded to them. Changes are marked with track changes in the manuscript.
Yours sincerely,
Wai Phyo Aung
Point 1: 46 Spell out non-communicable diseases the first time it is used and bracket (NCD)s
Response 1: The words “non-communicable diseaseas” are addeded in line 46-47 now.
Point 2: 54-5 As NCD is now to be spelled out in line 46, no need to do so here. Remove the bracket from NCD.
Response 2: We made changes accordingly in line 53-54.
Point 3: 73 Spell out fasting plasma glucose followed by bracketed (FPG).
Response 3: We had spelled out the word “fasting plasma glucose” in line 40. Therefore, the abbreviation was used in line 74.
Point 4: 200 Substitute “rise” for “raise”
Response 4: We updated the word “rise” accordingly in line 201.
Point 5: 205 Make “increase” plural by adding an “s” at the end.
Response 5: We made the changes according to the comment in line 206.
Point 6: 211 Add the article “the” “before” “disadvantages”
Response 6: We added the article “the” before “disadvantages” in line 212.
Point 7: 216 Each of “male” and “females” in this line should be made plural by adding an “s” at the end.
Response 7: We exchanged the words “men” and “women” instead of “male” and “female” in line 217.
Point 8: 220 “Male” and “Female” in this line should be made plural by adding an “s” at the end.
Response 8: We also exchanged the words “men” and “women” instead of “male” and “female” in line 221.
Point 9: 234 Make “role” plural by adding an “s” at the end.
Response 9: We updated the word “roles” by adding the “s” in line 235.
Point 10: 235 Add the article “the” before “breadwinners”.
Response 10: We added the article “the” before “breadwinners” in line 236.
Point 11: 236 Bracket (such as child raising). Substitute “fewer” for “less”.
Response 11: We made the changes according to your comments in line 237.
Point 12: 250 Eliminate “-“.
Response 12: We eliminated “-“ in line 251.
Point 13: 253 Eliminate “,”.
Response 13: We eliminated “,” in line 254.
Point 14: The conclusion should also explain the value in doing the research and how it might helpful for future research.
Response 14: Thank you for your comments. We have reported the valuable results in conclusion briefly in line 267-268 as “Studies on prevalence of DM and factors associated with risk will enable policy makers to initiate timely and efficient preventive measures”. Further, we added the information how to helpful to policy maker in line 271-272 as “This study suggest a focus on management of diabetes, including the use of traditional medicine, and on gender differences in future research”.
Point 15: 287 Remove the semi-colon before 2017.
Response 15: We removed the semi-colon in line 291.
Point 16: 333 Eliminate duplicated “2004”.
Response 16: We made the changes accordingly in line 336.
Point 17: 346 Eliminate duplicated “2018”.
Response 17: We removed the duplicated word “2018” in line 349.
Point 18: 347 Eliminated space before “Kim”
Response 18: The space was removed accordingly in line 350.
Point 19: 352 Eliminate two spaces before “Ministry”.
Response 19: The space was removed accordingly in line 355.
Point 20: 373 Change hyphen to dash.
Response 20: We changed the dash from hyphen in line 376.
Point 21: 375 Eliminate space before “Latt”.
Response 21: We removed the space in line 378.